# Bi-Functionalized Transferrin@MoS_2_-PEG Nanosheets for Improving Cellular Uptake in HepG2 Cells

**DOI:** 10.3390/ma16062277

**Published:** 2023-03-12

**Authors:** Si Xu, Shanshan Liang, Bing Wang, Jiali Wang, Meng Wang, Lingna Zheng, Hao Fang, Tingfeng Zhang, Yi Bi, Weiyue Feng

**Affiliations:** 1School of Pharmacy, Key Laboratory of Molecular Pharmacology and Drug Evaluation (Yantai University), Ministry of Education, Collaborative Innovation Center of Advanced Drug Delivery System and Biotech Drugs in Universities of Shandong, Yantai University, Yantai 264005, China; 2CAS Key Laboratory for Biomedical Effects of Nanomaterials and Nanosafety, Institute of High Energy Physics, Chinese Academy of Sciences, Beijing 100049, China; 3University of Chinese Academy of Sciences, Beijing 100049, China

**Keywords:** bi-functionalization, transferrin@MoS_2_-PEG nanosheets, HepG2 cellular uptake

## Abstract

Pre-coating with a protein corona on the surface of nanomaterials (NMs) is an important strategy for reducing non-specific serum protein absorption while maintaining targeting specificity. Here, we present lipoic acid-terminated polyethylene glycol and transferrin bi-functionalized MoS_2_ nanosheets (Tf@MoS_2_-PEG NSs) as a feasible approach to enhance cellular uptake. Tf@MoS_2_-PEG NSs can maintain good dispersion stability in cell culture medium and effectively protect MoS_2_ NSs from oxidation in ambient aqueous conditions. Competitive adsorption experiments indicate that transferrin was more prone to bind MoS_2_ NSs than bovine serum albumin (BSA). It is noteworthy that single HepG2 cell uptake of Tf@MoS_2_-PEG presented a heterogeneous distribution pattern, and the cellular uptake amount spanned a broader range (from 0.4 fg to 2.4 fg). Comparatively, the intracellular Mo masses in HepG2 cells treated with BSA@MoS_2_-PEG and MoS_2_-PEG showed narrower distribution, indicating homogeneous uptake in the single HepG2 cells. Over 5% of HepG2 cells presented uptake of the Tf@MoS_2_-PEG over 1.2 fg of Mo, about three-fold that of BSA@MoS_2_-PEG (0.4 fg of Mo). Overall, this work suggests that Tf coating enhances the cellular uptake of MoS_2_ NSs and is a promising strategy for improving the intracellular uptake efficiency of cancer cells.

## 1. Introduction

It has been well known that the biomolecular corona formed on particles upon contact with cell culture medium in vitro or human plasma in vivo plays a critical role in the efficacy of nanomedicines [1,2]. The formation of protein corona can impact the biological processes of NMs, such as long circulation time, biodistribution, immune cell activation, cytotoxicity, specific cellular uptake, targeting efficiency, and more [3]. For instance, the opsonin proteins in the biomolecular corona, including immunoglobulins and complement proteins, facilitate phagocytic recognition, leading to sequestration and reduced bioavailability of the NMs [4]. To ensure the efficacy and safety of nanomedicine, various strategies have been developed [5,6] to limit the non-specific protein adsorption on NMs, such as stealth surface modification on NMs with hydrophilic or zwitterionic polymers such as antifouling polyethylene glycol, poly(phosphoester) (PPE), zwitterionic poly(ethyl ethylene phosphate) (PEEP), poly(2-ethyl-2-oxazoline), etc. [5,6,7,8]. However, it has been demonstrated that even with antifouling polymer coatings, the non-specific protein adsorption on the surface of NMs is inevitable [9]. Recent studies have shown that precoating using endogenous biomolecules, such as albumin, clusterin, and fibrinogen, has emerged as a promising strategy [10,11]. Preformed albumin–Au nanorod complexes have been demonstrated to reduce cytotoxicity and macrophage uptake, promote biostability and long circulation time, and enhance tumor targeting and penetration [12]. The specific protein coating on the surface of NMs will influence their cellular uptake and modify the interaction among NMs themselves, which ultimately affects their intracellular fate and in vivo biodistribution.

Transition metal dichalcogenide NMs have gained significant attention in biomedical applications due to their desirable optical properties, large surface area, high surface free energy, and ultrathin structure [13]. Some proteins have been found to interact strongly with transition metal dichalcogenide NMs [3,14,15,16], such as human serum albumin (HSA), transferrin (Tf), and immunoglobulin G (IgG). For example, HSA may strongly adsorb onto the surface of MoS_2_ via their hydrophobic interaction between nonpolar benzene rings and disulfide bonds on MoS_2_ layers. Fibrinogen (Fg) and immunoglobulin G (IgG) have been found to increase cellular uptake by macrophages and induce strong pro-inflammatory responses due to the recognition of an NSs-IgG complex by Fc gamma receptors [3]. However, there is limited research on the successful pre-coating of molybdenum disulfide with proteins to confer targeting specificity.

Transferrin (Tf) has been widely used to functionalize NMs as it can be specifically recognized and taken up by Tf receptors overexpressed on the surface of various tumor cells [17]. Further, the Tf protein contains 19 intra-chain disulfide bonds, which can provide great potential for physisorption on the surface of MoS_2_ NSs via electrostatic interaction [18].

In this study, water-soluble MoS_2_ NSs were functionalized by lipoic acid-terminated polyethylene glycol (LA-PEG) and Tf to improve their colloidal stability, biocompatibility, and cellular uptake efficiency. To investigate the competitive binding role of Tf with SA on MoS_2_ sheets, MoS_2_ NSs were coated with PEG and BSA for comparison. The result showed that LA-PEG and Tf bi-functionalization improved MoS_2_ NSs stability and enhanced cellular uptake in HepG2 cells. This study provides a facile strategy for optimizing MoS_2_ NSs and new insights into their physical and chemical properties for their biomedical applications.

## 2. Experimental Section

### 2.1. Materials

Molybdenum disulfide dispersion (XF138) was obtained from Nanjing xfnano Material Technology Co., Ltd. (Nanjing, China) Lipoic acid-terminated polyethylene glycol amine (NH_2_-PEG-LA) (MW ~5 K) was purchased from Aladdin Co., Ltd. (Fukuoka, Japan) Bovine serum albumin and transferrin were purchased from Macklin Biochemical Co., Ltd. (Shanghai, China). Ultrapure water (Milli-Q, resistivity 18.2 MΩ cm) was used throughout the work.

### 2.2. Preparation and Characterization of TF@MoS_2_-PEG and BSA@MoS_2_-PEG

In order to synthesize TF@MoS_2_-PEG and BSA@MoS_2_-PEG, the stable dispersed PEG functionalized MoS_2_ nanosheets were first synthesized. Few-layered MoS_2_ nanosheets were obtained through bath sonication of MoS_2_ bulk sheets solution for 10 h with an output power of 585 W. The resulting dispersion was centrifuged at 10,000 rpm for 10 min to remove the bulk MoS_2_ sheets. The supernatant containing the exfoliated layers of MoS_2_ NSs was collected and subsequently mixed with LA-PEG (w:w: 1:10) under stirring overnight. The mixed suspension was centrifugated at 14,600 rpm for three times using 100 kDa MWCO filters (Millipore, Burlington, MA, USA) to remove excess LA-PEG-NH_2_ molecules. The obtained MoS_2_-PEG NSs were resuspended in ultrapure water and stored at 4 °C for future use. Further protein functionalization of Tf@MoS_2_-PEG or BSA@MoS_2_-PEG was prepared by incubation of 0.5 mL of 0.1 mg/mL MoS_2_-PEG NSs in 1 mg/mL of Tf or BSA solution (in 10 mM PBS, Beyotime Biotechnology Co. Ltd, Shanghai, China) at 37 °C, 120 rpm for 1 h. The final complex of TF@MoS_2_-PEG or BSA@MoS_2_-PEG was obtained via centrifugation at 14,000 rpm for 30 min to remove the supernatant, followed by resuspension in fresh PBS.

### 2.3. Physicochemical Characterization of MoS_2_-PEG, Tf@MoS_2_-PEG, and BSA@MoS_2_-PEG Complex

The physicochemical properties of PEG functionalized MoS_2_ nanosheets were characterized by atomic force microscopy (AFM), dynamic light scattering (DLS), and zeta potential analysis. AFM (AFM5500, HITACHI, Tokyo, Japan) was used to characterize the morphology and thickness of MoS_2_ NSs before and after PEG and Tf coating. 1.0 mg/mL samples were initially dispersed in an aqueous solution and then deposited onto a mica plate for AFM imaging. The height profile and lateral size of MoS_2_ nanosheets were analyzed by Nanoscope Analysis 2.0 software. Section analysis was used to measure the height of the MoS_2_ nanosheet. The AFM statistics were used to generate the lateral size distribution of MoS_2_ nanosheets. The hydrodynamic diameters and zeta potential of MoS_2_ nanosheets before and after PEG and Tf coating were measured based on DLS and electrophoretic light scattering techniques by a Malvern Mastersizer 2000 laser diffraction analyzer. The DLS analysis results include the average particle size and the polydispersity index (PDI), which is an indication of quality with respect to the size distribution.

The absorption of serum protein Tf or BSA on MoS_2_-PEG nanosheets was studied by the DLS and zeta potential analysis. The chemical state of Mo and S in MoS_2_-PEG NSs, Tf@MoS_2_-PEG, and BSA@MoS_2_-PEG was analyzed via synchrotron radiation X-ray photoelectron spectroscopy (SR-XPS). SR-XPS measurements were performed at the 4B9B beamline of Beijing Synchrotron Radiation Facility (BSRF, Beijing, China) equipped with a Si (111) double crystal monochromator. The core level spectra, including Mo 2p and S 2p, were recorded at the incident photon energy of 700 eV with a step size of 0.16 eV, using a hemispherical electron energy analyzer (HA150, VSW, Scienta, Uppsala, Sweden), with energy resolution better than 300 meV.

### 2.4. Competitive Adsorption of Tf with BSA on MoS_2_-PEG NSs

To further discern the Tf affinity to MoS_2_-PEG NSs, the exchange reaction of BSA@MoS_2_-PEG or Tf@MoS_2_-PEG in the presence of Tf or BSA was performed. A total of 0.5 mL of 0.1 mg/mL BSA@MoS_2_-PEG was incubated in 0.5 mL of Tf (1 mg/mL) at 37 °C, 120 rpm for 10 min, 30 min, and 60 min, respectively. In addition, the competitive adsorption of Tf and BSA on MoS_2_-PEG NSs was carried out in the co-presence of BSA and Tf. The protein-coated MoS_2_-PEG NSs were collected and resuspended in 500 μL of PBS after centrifugation at 14,000 rpm at 4 °C three times to remove the unreacted protein. The collected pellet was re-suspended in protein loading buffer (62.5 mM Tris-HCl pH 6.8, 2% (*w*/*v*) SDS, 10% glycerol, 0.04 M DTT, and 0.01% (*w*/*v*) bromophenol blue) for SDS-polyacrylamide gel electrophoresis (SDS-PAGE) analysis. The samples containing loading buffer were denatured at 100 °C for 5 min, and the supernatant was collected after centrifugation at 14,000 rpm for 10 min to remove MoS_2_ NSs. An amount of 10 μL of protein sample was loaded in 12%polyacrylamide gel, and proteins were separated on a 10% SDS-PAGE gel. Gel electrophoresis was performed at 120 V and 400 mA for about 60 min until the dye reached the bottom of the gel. Proteins were visualized by staining with Coomassie brilliant blue R-250 and de-staining overnight with a destaining solution (50% methanol, 10% acetic acid). The de-stained gel was scanned using UMAX UTA-2100XL-USB scanner. The band intensity was quantified by Image J software, and the data were normalized to the intensity of a fixed band.

### 2.5. Single Cellular Uptake Analysis

Human liver cancer cell (HepG2) was purchased from Cell Culture Center, Institute of Basic Medical Sciences of Chinese Academy of Medical Sciences (Beijing, China) and used for cellular uptake investigation. HepG2 cells were cultured in DMEM medium supplemented with 10% FBS and 1% penicillin-streptomycin solution at 37 °C and 5% CO_2_. Considering the biomedical application of Transferrin@MoS_2_-PEG, we evaluated the potential cytotoxic effects first. The dose- and time-dependent change of Tf@MoS_2_-PEG in HepG2 cell viability were performed. For cell viability analysis, after being cultured in 96-well plates for 12 h with the initial density of 2 × 10^4^ cells/well, the cells were treated with transferrin@MoS_2_-PEG at the doses of 1, 5, 25, and 125 μg/mL for 6, 12, and 24 h, while the control cells were treated with phosphate buffer solution. The cell viability was measured by a Cell Counting-8 Kit (CCK-8).

For a single-cell uptake analysis, the cells were seeded in 6-well plates at a density of 4 × 10^5^ cells/well, cultured overnight, and then exposed to 10 ppm of MoS_2_-PEG, BSA@MoS_2_-PEG, and Tf@MoS_2_-PEG at 1 h, 4 h, and 12 h. After treatment, cells were collected via digestion with trypsin EDTA, centrifuged at 1200 rpm for 3 min, and washed with 0.9% NaCl solution 3 times. The collected cells were immobilized with methanol at 4 °C, centrifuged to remove methanol, and resuspended in ultrapure water with a cell number density of 10^5^/mL. The ^95^Mo content in single cells was determined by inductively coupled plasma mass spectrometry (ICP-MS, NexION 300 D, Norwalk, CT, USA). Before analysis, ICP-MS was tuned using a 10 mg/L Mo-element standard solution. The dwell time was optimized to 4 ms and with a sample rate of 40 μL/min. The data were processed via Syngistix™ software to obtain the single-cell signal based on an iterative algorithm. The detailed data analysis was performed according to our previous description [19,20]. The temporal profile of ^95^Mo in a single HepG2 cell is directly related to the number of cells, and the signal intensity of a single-cell event was related to the Mo mass in one individual cell. Box plots were used to visualize the distribution patterns of Mo mass in cells.

## 3. Results and Discussion

### 3.1. Formation of Tf@MoS_2_-PEG Complex

The synthesis of Tf@MoS_2_-PEG NSs is illustrated in Figure 1A. Initially, the MoS_2_ nanosheets were synthesized by an ultra-sonication exfoliation method. Then, LA-PEG-NH_2_ polymer was utilized to functionalize MoS_2_ nanosheets through -S–S bond binding. Tf@MoS_2_-PEG was prepared by incubating Tf with PEG-functionalized MoS_2_ NSs. The typical AFM topography images of MoS_2_ bulk sheets, MoS_2_ nanosheets, MoS_2_-PEG, and Tf@ MoS_2_-PEG NSs are shown in Figure 1B. The primary MoS_2_ bulk sheets presented an average lateral dimension of 296 nm and an average thickness of 2.8 nm. After sonification, MoS_2_ NSs exhibited layered structures with an average lateral dimension of 110 nm and an average thickness of 2 nm. After PEGylation, the average lateral size of MoS_2_-PEG nanosheets did not show significant changes (~110 nm), while their average thickness increased to 5 nm, indicating that the PEG molecules attached on the surface of MoS_2_ nanosheets [21]. Further, Tf@MoS_2_-PEG NSs with a narrow lateral size distribution ranging from 50 to 170 nm and the thickness increased to 10 nm, also demonstrating Tf adsorption on MoS_2_-PEG NSs. The zeta potential of MoS_2_ NSs was −51.2 mV in water and decreased to −22.1 mV and −6.52 mV, in MoS_2_-PEG and Tf@MoS_2_-PEG NSs, respectively (Figure 1C), due to the charge shielding effect of the incorporated PEG layer [22]. The increase in the thickness and the drop in the zeta-potential absolute value indicated the Tf coating on MoS_2_-PEG NSs [23,24]. The stability of the MoS_2_ nanosheets after PEGylation in PBS was determined by DLS measurement (Figure 1D,E). The average hydrodynamic (HD) size of the MoS_2_-PEG NSs and Tf@MoS_2_-PEG NSs in PBS showed no obvious change from 1 h to 12 h, indicating their good dispersion stability in PBS solution. In PBS medium, the HD size of MoS_2_-PEG NSs and Tf@MoS_2_-PEG NSs was about 86~110 nm and ~260 nm, respectively, as shown in Figure 1D.

### 3.2. Competitive Adsorption of Tf and BSA on MoS_2_-PEG NSs

The DLS analysis showed that the average HD size of MoS_2_-PEG and Tf@MoS_2_-PEG was ~292 and ~280 nm in DMEM media and presented no obvious change within 12 h, indicating good dispersion stability (Figure 2A). On the other hand, the average HD size BSA@MoS_2_-PEG in DMEM solution presented two-modal distribution, higher PDI value, and gradually increased size within 12 h. About 75% of BSA@MoS_2_-PEG showed an HD size of 890 nm at 1 h, while 60% and 72% of BSA@MoS_2_-PEG showed HD size of ~4800 nm and 5200 nm at 4 h and 12 h, indicating significant agglomeration (Figure 2B). MoS_2_-PEG NSs exhibited a zeta potential of −34.3 mV in the DMEM medium. After Tf or BSA coating, their zeta potentials were about −24.8 and −15.4 mV in the DMEM medium.

To examine the stability of Tf@MoS_2_-PEG NSs in the physiological system, the interactions of Tf with BSA-precoated MoS_2_-PEG (BSA@MoS_2_-PEG) and the competitive adsorption of Tf and BSA on MoS_2_-PEG MoS_2_ NSs were investigated. Figure 2C shows that upon incubation with Tf-containing solution, BSA@MoS_2_-PEG showed a decrease in the intensity of the 66 kDa band (BSA molecular weight) with an increase in incubation time, while the intensity of the 80 kDa band (Tf molecular weight) showed an obvious increase. The quantitative analysis showed that Tf increased by about 1.3-fold of BSA content (Figure 2D), implying major exchange adsorption of Tf for BSA. Furthermore, upon co-incubation of MoS_2_ NSs with Tf and BSA, SDS-PAGE presented a prominent intense band at 80 kDa and a weak band at 66 kDa. The quantitative analysis showed that Tf content adsorbed on MoS_2_-PEG NSs was about 1.5-fold of BSA content, indicating a stronger affinity of Tf to MoS_2_-PEG NSs than BSA. These results indicated that MoS_2_ NSs exhibited preferential binding toward Tf relative to BSA.

### 3.3. Chemical State Analysis of MoS_2_ NSs before and after Protein Coating by Synchrotron Radiation XPS

Figure 3A shows the high-resolution Mo3d and S2p spectra of the pristine MoS_2_, MoS_2_-PEG, Tf@MoS_2_-PEG, and BSA@MoS_2_-PEG. The Mo spectra of all samples were deconvoluted into Mo(IV), Mo(V), and Mo(VI) spin-orbit doublet characteristic peaks. The binding energy of Mo3d_5/2_ (~228.0 eV) and Mo3d_3/2_ (~228.9 eV) was ascribed to the Mo(IV) state of MoS_2_ NSs [25]. The additional doublets were assigned to Mo(V) (~231.2 eV and ~233.4 eV) and Mo(VI) (~232.7 eV and ~235.9 eV), respectively. Under ultra-sonication, Mo(IV) may be oxidized to Mo(V) intermediate chemical form, which may be further oxidized into Mo(VI) in the presence of oxygen. Therefore, the XPS results indicated that some Mo(IV) on MoS_2_ NSs was oxidized to intermediate Mo(V) oxides and Mo(VI) oxides, leading to the formation of molybdenum oxide (MoO_3_) on the MoS_2_ surface [26]. The high-resolution S2p spectra of all the samples include spin-orbit doublets related to the S^2−^ (~162.1 eV and ~163.5 eV) and SO_x_^y−^ group (167.4 eV and 169.0 eV) [25,27]. Moreover, the formation of the SO_x_^y−^ group could be caused by local oxidation during the reaction. The quantitative atomic percentages of Mo and S in all the samples are given in Figure 3B. Furthermore, the Mo 3d and S2p XPS spectra in PEG functionalized MoS_2_ displayed no obvious differences from that of MoS_2_, except that the content of Mo(V) slightly decreased by ~4% and the content of Mo(IV) slightly increased by ~4%, indicating that part of Mo(V) reduced to Mo(IV) after PEG binding. Particularly, the content of Mo(IV), Mo(V), and Mo(VI) state in Tf@MoS_2_-PEG is 34.1%, 28.6%, and 37.3%, respectively, which is similar to those in MoS_2_-PEG (34.7%, 27.4%, and 37.9%, respectively), indicating Tf and PEG co-coating has no obvious effects on the chemical state of Mo in MoS_2_ NSs. The content of S^2−^ and SO_x_^y−^ in Tf@MoS_2_-PEG is 60.4% and 39.6%, respectively, which is similar to those in MoS_2_ NSs (62.5% and 37.5%, respectively). However, in the case of BSA coating, the content of S^2−^ decreased by 40.7%, and SO_x_^y−^ content significantly increased by 40.7%. Previous studies demonstrated that benzene rings and disulfides of BSA can bind more strongly to MoS_2_ nanosheets via strong hydrophobic and physical adsorption interaction as compared to the polar groups, such as carboxyl, amino, thiol, etc. [14]. Thus, BSA may induce layer-by-layer exfoliation of MoS_2_ NSs and get a more active site of S^2−^ oxidized into SO_x_^y−^. These studies indicated that Tf and PEG bi-functionalization could effectively protect MoS_2_ NSs from oxidation in ambient aqueous conditions.

### 3.4. Single-Cell Uptake Analysis by Time-Resolved ICP-MS

Single cellular uptake of MoS_2_ NSs is shown in Figure 4. Figure 4A shows the effects of the Tf@MoS_2_-PEG dose on HepG2 cellular viability and the morphology of HepG2 at the selected dose for single cellular uptake. Transferrin@MoS_2_-PEG showed a significant decrease in cell viability at the dose of 125 ppm for 24 h. A median dose of 10 ppm was selected for single cellular uptake analysis. At this concentration, the HepG2 cells maintained their integrity and mono-dispersed state. Figure 4B presents the temporal profile of ^95^Mo in HepG2 cells (2 × 10^5^ cells/mL) by ICP-MS. The data showed that after treatment of Tf@MoS_2_-PEG, BSA@MoS_2_-PEG, and MoS_2_-PEG, the number of single-cell events for uptake of MoS_2_ NSs gradually increased with the exposure time from 1 h to 12 h. Figure 4C shows that the ingested average mass of HepG2 cells follows the order of Tf@MoS_2_-PEG>BSA@MoS_2_-PEG>MoS_2_-PEG. The average uptake mass of Mo in HepG2 cells was treated with MoS_2_-PEG is 0.03 fg, 0.12 fg, and 0.12 fg at 1 h, 4 h, and 12 h, respectively, while the ingested average Mo amount in HepG2 cells was treated with Tf@MoS_2_-PEG is 0.35 fg, 0.40 fg, and 0.41 fg at 1 h, 4 h, and 12 h, respectively, which was higher than the uptake amount of Mo in those cells treated with BSA@MoS_2_-PEG (0.10 fg, 0.15 fg, 0.41 fg at 1 h, 4 h, and 12 h, respectively). Particularly, it is noteworthy that the masses of Mo in single HepG2 cells treated with Tf@MoS_2_-PEG span a broader range (from 0.4 fg to 2.4 fg), demonstrating larger variations in a population of HepG2 cells. Further, over 5% of HepG2 cells presented uptake of the Tf@MoS_2_-PEG over 1.2 fg of Mo, about three-fold that of BSA@MoS_2_-PEG. Comparatively, the masses of Mo in HepG2 cells treated with BSA@MoS_2_-PEG and MoS_2_-PEG showed narrower distribution, indicating homogeneous uptake in the single HepG2 cell.

Studies have shown that nanoparticle-protein conjugates are attractive for targeting intracellular delivery [28]. However, most experimental evidence shows that direct protein coating on NMs is prone to inducing the aggregation of NMs, changing the original molecular identity of the nanocarrier system [29]. In addition, direct protein coating of NMs has the propensity for the formation of new corona in a biomolecular environment, which could mask the targeting ligand, resulting in the loss of targeting ability and compromising the therapeutic efficacy [9]. Therefore, it is critical and necessary to develop an efficient and versatile nanocarrier system for targeted delivery to the desired cell. In this study, the surface modification of the MoS_2_ NSs with PEG and Tf was used to enhance the colloidal stability and intracellular uptake. This study provides a facile strategy to improve the delivery efficiency of target cancer cells.

## 4. Conclusions

In summary, we utilized LA-PEG and Tf to functionalize MoS_2_ nanosheets, which improved their colloidal stability and intracellular delivery efficiency. MoS_2_-PEG NSs demonstrated a higher affinity toward Tf compared to BSA. Additionally, co-functionalization using Tf and PEG resulted in MoS_2_ NSs with excellent colloidal stability in various conditions, including cell culture medium, and could protect MoS_2_ NSs from oxidative degradation. It is noteworthy that single HepG2 cell uptake of Tf@MoS_2_-PEG exhibited a heterogeneous distribution pattern and a broader range of cellular uptake amounts ranging from 0.4 fg to 2.4 fg. Importantly, over 5% of HepG2 cells showed uptake of the Tf@MoS_2_-PEG over 1.2 fg of Mo, about three-fold that of BSA@MoS_2_-PEG (0.4 fg of Mo). This work suggests that Tf coating could enhance cellular uptake of MoS_2_ NSs and is a promising strategy to improve the delivery efficiency to target cancer cells.

## Figures and Tables

**Figure 1 materials-16-02277-f001:**
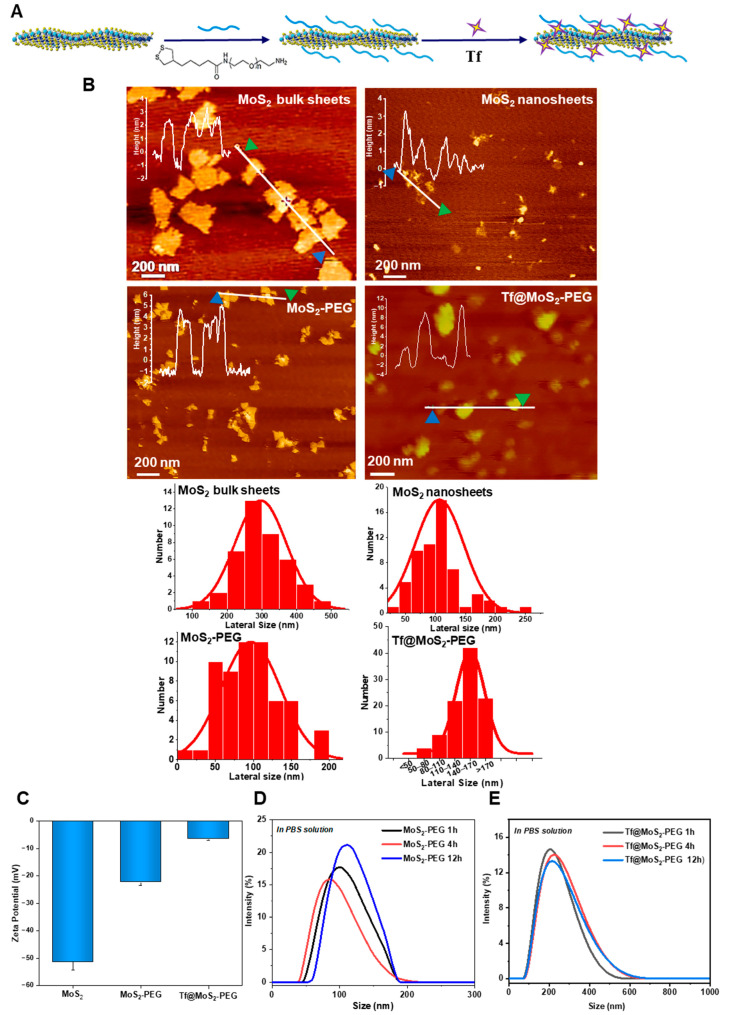
(**A**) Schematic illustration of the synthesis of Tf@MoS_2_-PEG nanosheets. (**B**) AFM images of MoS_2_ bulk sheets before ultrasonication, MoS_2_ nanosheets, MoS_2_-PEG, and Tf@MoS_2_-PEG nanosheets. The insert figures indicate the height measurements along with the line profiles. The histograms show the lateral size distribution of MoS_2_ bulk sheets, MoS_2_ nanosheets, MoS_2_-PEG, and Tf@MoS_2_-PEG NSs. (**C**) Zeta potential of MoS_2_, MoS_2_-PEG, and Tf@MoS_2_-PEG in deionized water. The hydrodynamic diameters of MoS_2_-PEG (**D**) and Tf@ MoS_2_-PEG (**E**) in PBS solution.

**Figure 2 materials-16-02277-f002:**
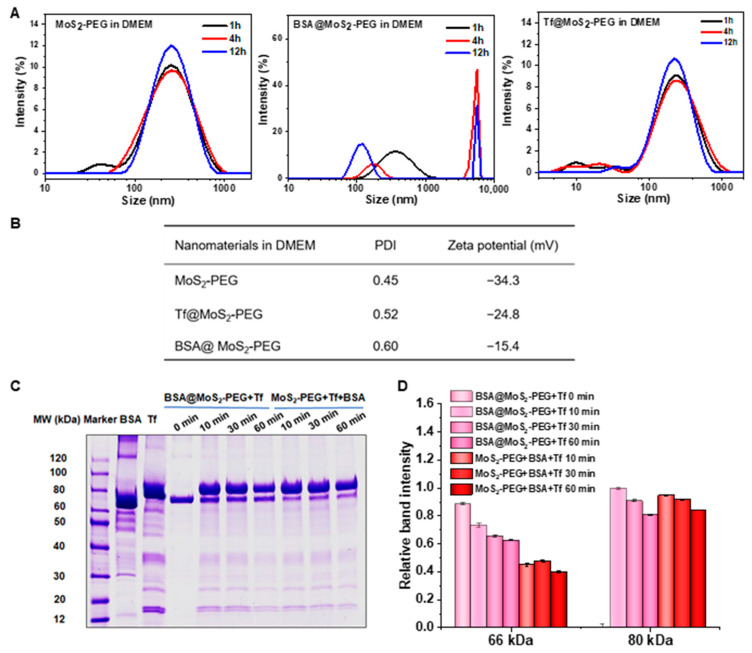
(**A**) Hydrodynamic size of MoS_2_-PEG, Tf@MoS_2_-PEG, and BSA@MoS_2_-PEG in DMEM. (**B**) Polydispersity index (PDI) and zeta potential of MoS_2_-PEG, Tf@MoS_2_-PEG, and BSA@MoS_2_-PEG in DMEM. (**C**) SDS-PAGE analysis of BSA@MoS_2_-PEG in Tf and MoS_2_-PEG in BSA and Tf solutions. (**D**) Quantitative analysis of relative amounts of BSA and Tf binding on MoS_2_-PEG and BSA@MoS_2_-PEG nanosheets.

**Figure 3 materials-16-02277-f003:**
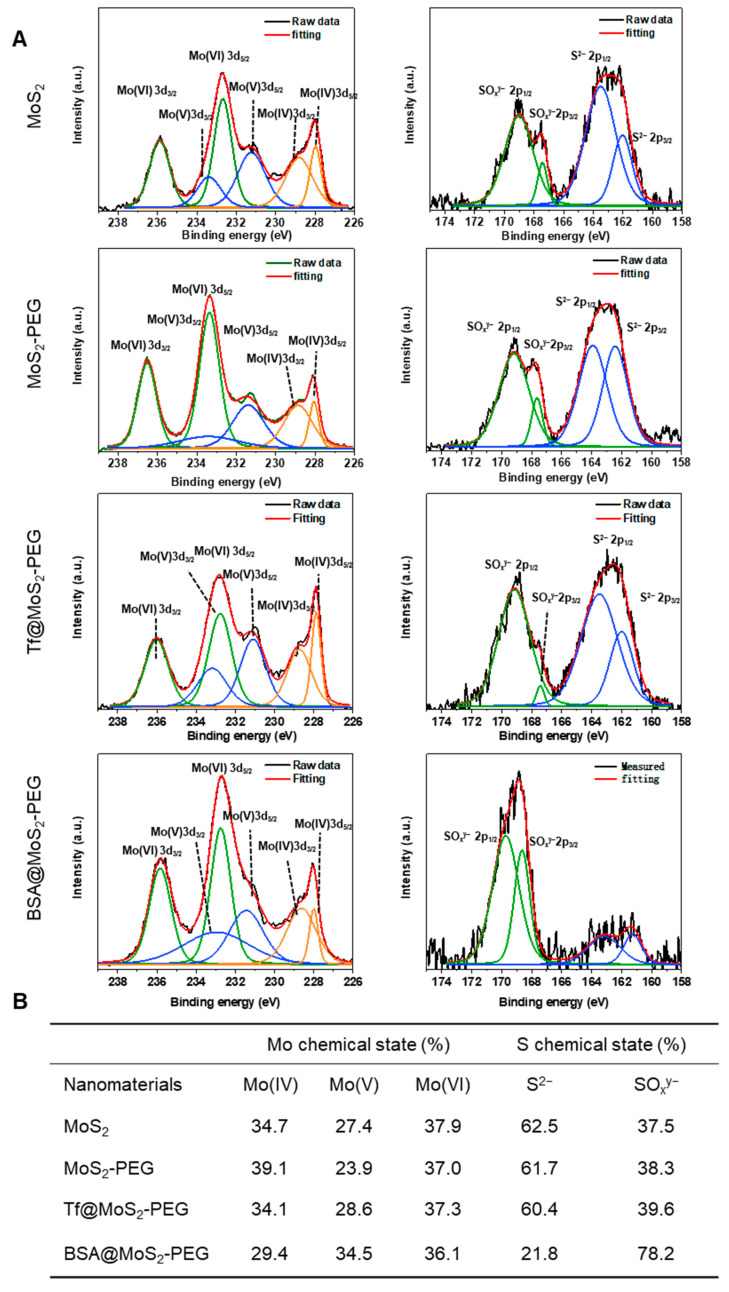
(**A**) High-resolution XPS spectra of Mo 3d and S 2p in MoS_2_, MoS_2_−PEG, Tf@ MoS_2_−PEG, and BSA@MoS_2_−PEG. (**B**) Quantitative results of Mo and S chemical states in MoS_2_, MoS_2_−PEG, Tf@ MoS_2_−PEG, and BSA@MoS_2_−PEG. The green, blue, and orange color spectra components were attributed to Mo(VI), Mo(V), and Mo(IV) species.

**Figure 4 materials-16-02277-f004:**
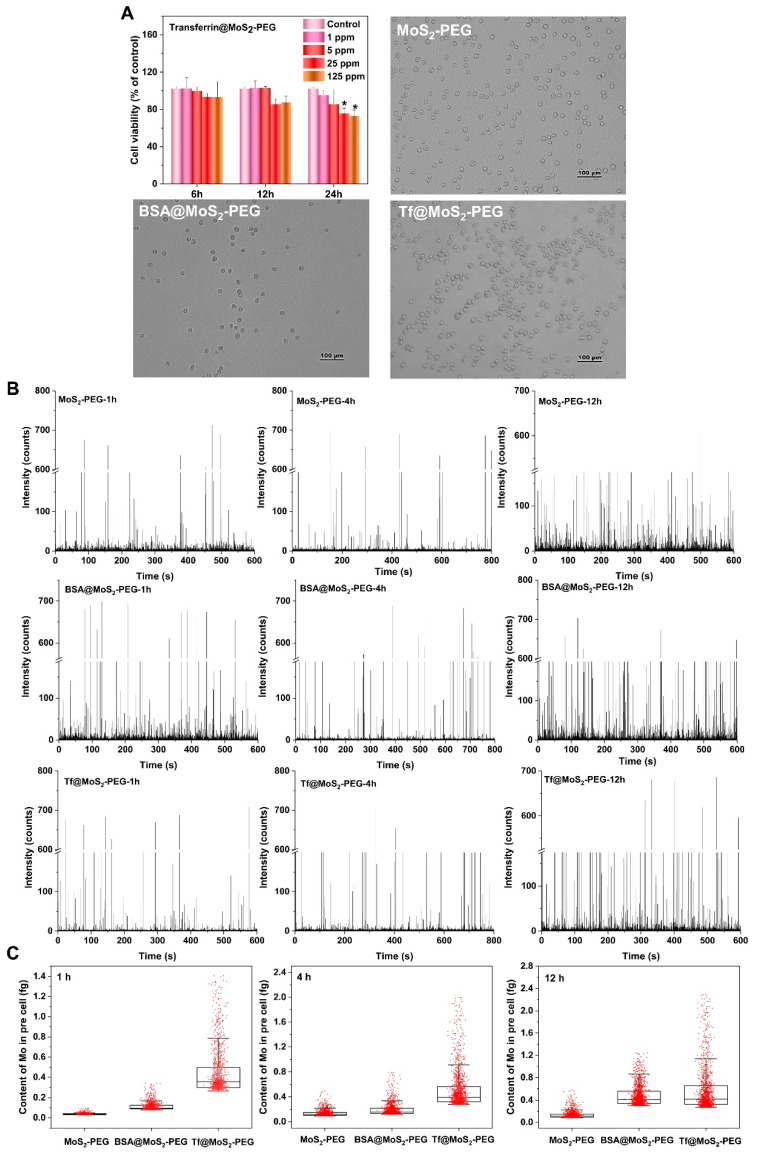
(**A**) Effect of transferrin@MoS_2_-PEG on cell viability after treatment with 1, 5, 25, and 125 μg/mL for 6, 12, and 24 h and microscopy images of intact and dispersed HepG2 cells after treatment with 10 μg/mL transferrin@MoS_2_-PEG for 12 h in ultrapure water after immobilization with 70% *v*/*v* methanol. * *p* < 0.05 vs. the control. (**B**) The spectra of SC-ICP-MS measurement of ^95^Mo in individual HeLa cells (cell density = 2 × 10^5^ cells/mL). (**C**) Box plot for the contents of Mo in single HepG2 cell after treatment of 10 μg/mL MoS_2_-PEG, BSA@MoS_2_-PEG, and Tf@MoS_2_-PEG.

## Data Availability

Not applicable.

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
