# Peer review of "Bi-Functionalized Transferrin@MoS2-PEG Nanosheets for Improving Cellular Uptake in HepG2 Cells"

_materials, 2023, doi:10.3390/ma16062277_

Round 1

Reviewer 1 Report

Feng et al. described the preparation and characterization of bi-functionalized Transferrin decorated MoS2-PEG nanosheets for improving HepG2 uptake.

The obtained results are of interest even if, most of the time, it is complicated to follow the reasoning of the authors. The authors have to carefully read their manuscript in order to correct the English and all the typos (Upper case letter at the beginning of a word after a point, space between the point and the first word, etc.). Moreover, the authors have to be homogenous in their way to write numbers (10 000 rpm vs 14600 rpm vs 14,000 rpm): in English, the right way is the following: 14,000 rpm.

In view of these general comments and those given below, I do recommend the publication of this manuscript in Materials after major revision.

Please find below, my specific comments:

1. Page 1, Introduction: The sentence “Various strategies have been … protein adsorption on NMs.” Is not clear: Are various strategies been developed to limit or to ensure efficacy and safety of nanomedicines?

2. Page 1, Introduction: Is the sentence “such as stealth surface modification … poly(2-ethyl-2-oxazoline), etc5-8” a part of the previous one? I guess, yes. In this case, the authors have to change the point to a comma.

3. Page 2, Introduction line 11: What does “SA” mean? The authors have to define all the abbreviations they used.

4. Page 2, Introduction line 21: What does “Physisorption” mean?

5. Page 3, Experimental part line 4: The two abbreviations “AFM” and “DLS” have to be defined. Moreover, more details have to be given on these analyses. How were prepared the samples for AFM and how were they analyzed? What about the method used to measure the diameter and zeta potential? What about the samples’ dispersity (PDI)?

6. Page 4, Experimental part line 2: “… in single HeLa cells …”, Are the authors worked with HeLa or HepG2 cells?

7. Page 4, Results and Discussion Fig. 1B: I suggest to the authors to put the histograms outside of the AFM images because it is impossible to see these images correctly. Moreover, reading these histograms is difficult or impossible.

8. Page 4, Results and Discussion line 27: The Fig. 1D gave the HD size in PBS not in DMEM medium. Please check and correct.

9. Page 4, Results and Discussion: I think that a part of the sentence “Fig. 2C shows upon … “  is missing.

10. Page 4, Results and Discussion: The sentence “The quantitative analysis … NSs than BSA.” Is not clear and needs to be reworded.

11. Page 6 Results and Discussion: I guess that a part of the sentence “In the study, surface … and intracellular uptake.” is missing.

Author Response

Review 1

Feng et al. described the preparation and characterization of bi-functionalized Transferrin decorated MoS2-PEG nanosheets for improving HepG2 uptake.

The obtained results are of interest even if, most of the time, it is complicated to follow the reasoning of the authors. The authors have to carefully read their manuscript in order to correct the English and all the typos (Upper case letter at the beginning of a word after a point, space between the point and the first word, etc.). Moreover, the authors have to be homogenous in their way to write numbers (10 000 rpm vs14600 rpm vs14,000 rpm): in English, the right way is the following: 14,000 rpm.

In view of these general comments and those given below, I do recommend the publication of this manuscript in Materials after major revision.

Please find below, my specific comments:

  1. Page 1, Introduction: The sentence “Various strategies have been … protein adsorption on NMs.” Is not clear: Are various strategies been developed to limit or to ensure efficacy and safety of nanomedicines?

Reply: Thanks you for your suggestion. The expression “Various strategies have been … protein adsorption on NMs.” has been revised as “To ensure efficacy and safety of nanomedicine, various strategies have been developed to limit the non-specific protein adsorption on NMs” in page 1, the first paragraph.

  1. Page 1, Introduction: Is the sentence “such as stealth surface modification … poly(2-ethyl-2-oxazoline), etc5-8” a part of the previous one? I guess, yes. In this case, the authors have to change the point to a comma.

Reply: Thanks you for your suggestion. The point has been revised as a comma.

  1. Page 2, Introduction line 11: What does “SA” mean? The authors have to define all the abbreviations they used.

Reply: Thank you for your suggestion. The abbreviation “SA” has been revised as “HSA” in page 2, line 11.

  1. Page 2, Introduction line 21: What does “Physisorption” mean?

Reply:Thank you for your suggestion. The meaning “electrostatic interaction” of “Physisorption” interaction has been supplemented in the end of this expression in page 2, line 21.

  1. Page 3, Experimental part line 4: The two abbreviations “AFM” and “DLS” have to be defined. Moreover, more details have to be given on these analyses. How were prepared the samples for AFM and how were they analyzed? What about the method used to measure the diameter and zeta potential? What about the samples’ dispersity (PDI)?

Reply:Thank you for your suggestion. The “AFM” and “DLS” definition and related description of experimental details about sample preparation and analysis have been supplemented in page 3, the second paragraph.

  1. Page 4, Experimental part line 2: “… in single HeLa cells …”, Are the authors worked with HeLa or HepG2 cells?

Reply:Thank you for your suggestion. I am very sorry for the error. The “Hela” has been revised as “HepG2” cells in page 4, line 18.

  1. Page 4, Results and Discussion Fig. 1B: I suggest to the authors to put the histograms outside of the AFM images because it is impossible to see these images correctly. Moreover, reading these histograms is difficult or impossible.

Reply:Thank you for your suggestion. The histograms in Fig. 1B have been placed outside of the AFM images

  1. Page 4, Results and Discussion line 27: The Fig. 1D gave the HD size in PBS not in DMEM medium. Please check and correct.

Reply:Thank you for your suggestion. I am very sorry for the error. The expression “DMEM” has been corrected in page 4, line 48.

  1. Page 4, Results and Discussion: I think that a part of the sentence “Fig. 2C shows upon … “  is missing.

Reply:Thank you for your suggestion. The expression has been corrected.

  1. Page 4, Results and Discussion: The sentence “The quantitative analysis … NSs than BSA.” Is not clear and needs to be reworded.

Reply:Thank you for your suggestion. The expression “The quantitative analysis showed that Tf content is about 1.5-fold of BSA content, indicating Tf shows a stronger affinity to MoS2-PEG NSs than BSA” has been revised as “The quantitative analysis showed that Tf content adsorbed on MoS2-PEG NSs is about 1.5-fold of BSA content, indicating Tf shows a stronger affinity to MoS2-PEG NSs than BSA” in page 5, line 20~21

  1. Page 6 Results and Discussion: I guess that a part of the sentence “In the study, surface … and intracellular uptake.” is missing.

Reply:Thank you for your suggestion. I am very sorry for this error. The expression “In the study, surface … and intracellular uptake.” has been revised as “In this study, the surface modification of the MoS2 NSs with PEG and Tf was used to enhance the colloidal stability and intracellular uptake” in page 6, line 33.

We have completely checked the spell, formatting, punctuation, and grammar errors in the whole text.

We hope our revision of the manuscript could be satisfied and can be accepted for publication in Materials.

Best regards,

                                                                Sincerely yours

                                                                 Dr. Bing Wang

Reviewer 2 Report

The manuscript entitled “Bi-Functionalized Transferrin@MoS2-PEG Nanosheets for Improving Cellular Uptake in HepG2 Cells” describes the coating of molybdenum disulfide nanosheets with lipoic acid-PEG and the protein transferrin. After the physical-chemical characterization of the obtained material Transferrin@MoS2-PEG, its ability to be internalized into HepG2 cancer cells is explored by single-cell uptake analysis.

The manuscript explores an interesting topic, nevertheless, there are a few points to be addressed before publication.

1.         Considering the biomedical application of Transferrin@MoS2-PEG it is crucial to investigate the cell viability after incubation with the proposed system. In the reviewer’s opinion, it is necessary to perform toxicity/cytocompatibility studies on both cancer cells and normal cells (for example HepG2 and fibroblasts) to evaluate the potential effects on cell death/cell metabolic activity and, more importantly, to demonstrate the biomaterial safety on non-tumor cells.

2.       To make the 2D Figure clearer and more readable, it is suggested to use more distinguishable colors.

3.       The authors write that “Few-layered MoS2 nanosheets was obtained through bath sonication of MoS2 bulk sheets solution..”. Could you please provide further details about the size of the starting material? It could be desirable to better understand the potentialities of this method of production.

4.       Please, check spell, formatting, and punctuation in the whole text. Here are some examples: “. , Various strategies have been developed to limit to ensure efficacy and safety of nanomedicine through limiting the non-specific protein adsorption on NMs. such as stealth…”; “…subsequently mixed with LA-PEG (1:10) under stirring overnight.” is 1:10 a weight/weight ratio?; “..was loaded in 12% gel polyacrylamide gel..”

Author Response

Reviewer 2

The manuscript entitled “Bi-Functionalized Transferrin@MoS2-PEG Nanosheets for Improving Cellular Uptake in HepG2 Cells” describes the coating of molybdenum disulfide nanosheets with lipoic acid-PEG and the protein transferrin. After the physical-chemical characterization of the obtained material Transferrin@MoS2-PEG, its ability to be internalized into HepG2 cancer cells is explored by single-cell uptake analysis.

The manuscript explores an interesting topic, nevertheless, there are a few points to be addressed before publication.

  1. Considering the biomedical application of Transferrin@MoS2-PEG it is crucial to investigate the cell viability after incubation with the proposed system. In the reviewer’s opinion, it is necessary to perform toxicity/cytocompatibility studies on both cancer cells and normal cells (for example HepG2 and fibroblasts) to evaluate the potential effects on cell death/cell metabolic activity and, more importantly, to demonstrate the biomaterial safety on non-tumor cells.

Reply:Thank you for your suggestion. The dose and time-dependent cytotoxicity of transferrin@MoS2-PEG on HepG2 cells was evaluated. The related experimental details and the results have been added in page 4, the first paragraph and page 6, the first paragraph.

  1. To make the 2D Figure clearer and more readable, it is suggested to use more distinguishable colors.

Reply:Thank you for your suggestion. Fig.2D has been redrawn.

  1. The authors write that “Few-layered MoS2nanosheets was obtained through bath sonication of MoS2bulk sheets solution.” Could you please provide further details about the size of the starting material? It could be desirable to better understand the potentialities of this method of production.

Reply:Thank you for your suggestion. The AFM image of the starting material has been added in Fig.1B, and the related results has been added in page 4, line 30-33

  1. Please, check spell, formatting, and punctuation in the whole text. Here are some examples: “. , Various strategies have been developed to limit to ensure efficacy and safety of nanomedicine through limiting the non-specific protein adsorption on NMs. such as stealth…”; “…subsequently mixed with LA-PEG (1:10) under stirring overnight.” is 1:10 a weight/weight ratio?; “..was loaded in 12% gel polyacrylamide gel..”

Reply:Thank you for your suggestion. I am very sorry for these errors. The spell, formatting, and punctuation in the whole text has been completely checked and corrected. The expression “mixed with LA-PEG (1:10) under stirring overnight.” has been revised as “mixed with LA-PEG (W:W: 1:10) under stirring overnight”

We hope our revision of the manuscript could be satisfied and can be accepted for publication in Materials.

Best regards,

                                                                Sincerely yours

                                                                 Dr. Bing Wang

Round 2

Reviewer 1 Report

The authors have answered correctly to the questions raised by the reviewers and therefore improved their manuscript which can be published in Materials.

Reviewer 2 Report

In the reviewer's opinion, the modifications made by the authors improved the manuscript, making it ready for publication